# Mechanistic synergy of hair growth promotion by the Avicennia marina extract and its active constituent (avicequinone C) in dermal papilla cells isolated from androgenic alopecia patients

Woraanong Prugsakij[1], Sukanya Numsawat[2,3], Ponsawan Netchareonsirisuk[2,4], Parkpoom Tengamnuay[1], Wanchai De-Eknamkul[4,5]*

1 Department of Pharmaceutics and Industrial Pharmacy, Faculty of Faculty of Pharmaceutical Sciences, Chulalongkorn University, Bangkok, Thailand, 2 Department of Biochemistry and Microbiology, Faculty of Pharmaceutical Sciences, Chulalongkorn University, Bangkok, Thailand, 3 Bureau of Drug Control, Food and Drug Administration, Ministry of Public Health, Nonthaburi, Thailand, 4 Natural Product Biotechnology Research Unit, Faculty of Pharmaceutical Sciences, Chulalongkorn University, Bangkok, Thailand, 5 Department of Pharmacognosy and Pharmaceutical Botany, Faculty of Pharmaceutical Sciences, Chulalongkorn University, Bangkok, Thailand

* wanchai.d@chula.ac.th

## Abstract

Androgenic alopecia (AGA) is associated with an increased production of 5α-dihydrotestosterone (DHT) by steroid-5α-reductase (5α-R). Crude extracts from *Avicennia marina* (AM) and its active constituent, avicequinone C (AC), can inhibit 5α-R. We have, herein, explored the potential use of the AM extract and of AC as anti-AGA agents. To this end, we employed human dermal papilla cells (DPCs) isolated from AGA patients' hair that express 5α-R type-1 as well as the androgenic receptor (AR) at high levels. Our in vitro experiments revealed that the AM extract (10 μg/mL) and the AC (10 μM) exhibit multiple actions that interfere with the mechanism that causes AGA. Beside acting as 5α-R inhibitors, both preparations were able to inhibit either the DHT-AR complex formation or its translocation from the cytoplasm into the nucleus (the site of DHT's action). The treatments also increased the gene expression of growth factors in DPCs; these factors play important roles in the angiogenesis associated with hair growth. Moreover, the AM extract suppressed the apoptotic pathway, thereby postponing the initiation of the catagen phase. Taken together, our findings suggest that the AM extract and the AC could serve as natural sources for hair growth promotion and AGA treatment.

## Introduction

Scalp hair follicle undergoes cycles in order to produce new hairs throughout life. It is controlled by numerous factors such as genetics, hormones, cytokines, and a number of growth

**Data Availability Statement:** All relevant data are within the paper and its Supporting Information files.

**Funding:** PERDO's Center of Excellence on Medical Biotechnology (CEMB), MHESI; Chulalongkorn University's Research Unit for Natural Product Biotechnology (Grant No. GRU6101133003-1) and by The National Research Council of Thailand (Grant No. N72B650340).

**Competing interests:** The authors have declared that no competing interests exist.

factors [1]. Androgenic alopecia (AGA) is the most common type of hair loss in both men and women who have a genetical predisposition coupled with the overproduction of 5α-dihydro-testosterone (DHT) [2]. DHT is formed from the conversion of testosterone (T) by the action of the steroid-5α-reductase (5α-R) enzymes present in the cells of many organs, including the dermal papilla cells (DPCs) in the hair follicles [3]. Mechanistically, both T and the much more potent DHT can bind to the androgen receptor (AR), and form ligand-receptor complexes. The latter are then translocated into the nucleus and bind on the androgen response elements in the regions of the androgen-regulated genes, in order to initiate the signaling cascade in DPCs [4]. In AGA patients, the action of DHT affects the hair follicle by causing premature entering of the hair cells into the catagen phase (apoptosis and regression phase), thereby resulting in a shortening of the growing phase (anagen phase), and causing miniaturization events that change terminal hairs to non-growing vellus hairs [5]. Moreover, the overproduction of DHT has been reported to alter the production of some growth factors that are secreted by DPCs, including the insulin-like growth factor-1 (IGF-1), the fibroblast growth factor (FGF), the keratinocyte growth factor (KGF), the hepatocyte growth factor (HGF), and the vascular endothelial growth factor (VEGF); all known to be involved in the regulation of the hair growth cycle [6, 7]. Among these, HGF and VEGF have been found to be the main factors involved in angiogenesis, by inducing the formation of blood vessels within the hair follicles [8]. HGF has also been reported to regulate the hair follicle morphogenesis, including the cell proliferation, differentiation, and apoptosis [9]. Likewise, KGF (or FGF-7) is known to regulate the hair follicle proliferation and differentiation, to stimulate the hair fiber elongation in human scalp hair follicles, as well as to inhibit the hair cycle transition from the anagen to the catagen phase [10]. Furthermore, changes of some protein levels in the apoptotic pathway, including B-cell CLL/lymphoma 2 (Bcl-2), Bcl-2-associated protein (Bax), and cleaved caspase-3, have been shown to directly affect the catagen phase in DPCs [11, 12]. The increase in the expression of Bcl-2 (an antiapoptotic molecule) promotes the DPCs' survival, while the decrease in the expression of Bax and cleaved caspase-3 (pro-apoptotic proteins) stimulates cell survival [13]. It has also been reported that the hair shaft differentiation during the hair growth cycle is controlled by the bone morphogenic protein (BMP) pathway [14], while the induction of the BMP-4 expression in the BMP pathway significantly impedes the hair follicle proliferation [15].

To date, there are limitations in the use of finasteride (an orally administered drug) and of minoxidil (a topically-applied drug); the two FDA-approved synthetic drugs for treating AGA through different mechanisms of action. Finasteride is a selective inhibitor of 5α-R type 2 that decreases the serum DHT levels, while minoxidil is a potassium channel activator that modifies the cell membrane hyperpolarization and causes vascular muscle enlargement, thereby resulting in the regrowth of the hair follicle [16–18]. Finasteride has been reported to exert undesirable side effects, such as libido, abnormal erection, and gynecomastia [19, 20], while minoxidil and finasteride are characterized by a temporary efficacy [21, 22]. Therefore, searching for new substances that could prevent hair loss and stimulate hair growth in AGA, is still necessary.

Natural compounds are interesting due to their structural diversity, which is particularly important when considering them as a potential source for drug development. *Avicennia marina* (AM) is a mangrove species belonging to the family of Avicenniaceae. It has been used as a traditional medicine in Egypt in order to cure skin diseases [23]. AM has been shown to have many biological activities, namely antimicrobial, antioxidant, and antitumor activities [24, 25]. It can be the source of various compounds, such as terpenoids, anthraquinones, steroids, naphthalene, flavones, glucosides, and xanthones, that can be extracted from different parts of the plant (e.g. the barks, the leaves, the twigs, and its endophytes) [26]. Our previous

studies have shown that the crude extract of AM as well as its active constituent (avicequinone C; AC) can inhibit 5α-R [27, 28]. However, there has been no information regarding other actions of AM and AC that might possibly interfere with the biological processes causing AGA. Therefore, the aim of this study was to investigate the effects of both preparations on the expression of various hair growth factors (such as IGF-1, KGF, HGF, and VEGF), and their ability to inhibit the androgenic action of DHT through an interference at the levels of the DHT-AR complex formation or its translocation from the cytoplasm into the nucleus of the DPCs.

## Materials and methods

### Chemicals and reagents

All organic solvents were of analytical grade, and were purchased from RCI Labscan (Bangkok, Thailand). Paraformaldehyde (4%), finasteride, flutamide, toluidine blue, the trypan blue solution, the GENEzol™ reagent, bovine serum albumin (BSA), 0.2% TritonX-100, T, and DHT were purchased from Sigma-Aldrich (St. Louis, MO, USA). Dutasteride was purchased from BDG Synthesis (Wellington, New Zealand). Agarose-LE was purchased from Affymetrix (Santa Clara, CA, USA). The antibiotic-antimycotic solution (100×) and PrestoBlue® (10×) were purchased from Life Technologies (Carlsbad, CA, USA). Fetal bovine serum (FBS), 1640 Medium, Dulbecco's Modified Eagle Medium (DMEM), phosphate buffer solution (PBS), Williams' media E, Tris-acetate-EDTA buffer (50×), 0.25% trypsin-EDTA, the Platinum® Taq polymerase kit, the GeneRuler 1-kb DNA ladder DNase I enzyme, EDTA, first-strand cDNA synthesis kit, dATP, dTTP, dCTP, and dGTP were purchased from Thermo Fisher Scientific (Pittsburgh, PA, USA). The RNeasy® mini kit was purchased from Qiagen (Valencia, CA, USA), while the mesenchymal stem cell growth supplement was purchased from Science Cell Research Laboratories (Carlsbad, CA, USA). The monoclonal rabbit AR antibody was purchased from Cell Signaling Technology (Danvers, MA, USA), while the secondary Alexa Flur488-labeled donkey anti-rabbit IgG antibody was purchased from Invitrogen (Eugene, OR, USA).

### Plant material and active compound

The heartwood powder of AM was purchased from Chao Krom Poe Dispensary; a local Thai-Chinese medicinal store (Bangkok, Thailand) in June 2019 (Lot No. 190619). The raw material was determined to have an extractive value of 7.0%w/w using 95% ethanol and moisture content of 9.5%w/w by thermogravimetric method. The raw material was authenticated by comparing its TLC-chemical fingerprint of the ethanol extract with that reported previously [28]. AC was obtained by chemical synthesis, according to a method previously described in detail [29].

### Preparation of the AM extract and quality assessments

The AM raw material was first ground into powder, followed by an ethanol extraction using the method as described previously [28]. Practically, the extraction was conducted by maceration (200 g/L of 95% ethanol) for 24 h at room temperature for three cycles. The ethanolic extracts were pooled and evaporated until dryness at 40°C using a rotary evaporator. The quality assessment of the obtained AM extract was done by TLC which showed the chemical profile of the AM extract with AC used as a standard marker (S1A Fig in S1 File). The TLC system and conditions used in the analysis has been described in the report published previously [28]. The AM extract appeared to be relatively stable upon storage for at least three years at -20°C as

shown by the same TLC analysis. Only slightly decrease in the intensity of the AC band was observed (S1A Fig in S1 File). Quantitatively, a standard curve of AC marker (working range: 100–1000 ng with correlation coefficient ($r^2$) = 0.99) was constructed based on TLC densito-metric method using the wavelength of 254 nm (S1B Fig in S1 File). The results showed that the AM extract freshly prepared in June 2019 contained the AC content of 6.2% dry weight compared with 5.1% AC of the same AM extract stored for three years under -20˚C. The structure of AC in the AM extract was identified as avicequinone C based on the spectral data of both 1H-NMR and 13C-NMR reported previously [28]. In this study, the stock solutions of the AM extract (1 mg/mL) and of the AC compound (1 mM) were prepared by using DMSO and kept at 4˚C before use.

## Ethics statement

Human hair follicles were obtained from AGA patients during a hair transplantation surgery at the Million Hair Transplant Center (Bangkok, Thailand). The protocol was approved from the Ethics Review Committee for Research Involving Human Research Subjects of the Health Sciences Group of Chulalongkorn University, Bangkok, Thailand (Permit No. CU-220.2/59/2018). Male scalp follicles were obtained from five subjects with written informed consent. Minor subjects were not included in this study.

## Isolation of DPCs from human hair follicles, and cell culture establishment

DPCs were isolated from the collected hair samples by micro-dissection, and were cultured by using the method described previously [30, 31] with slight modification. Briefly, each hair follicle was dissected under the stereomicroscope at the hair root area of the dermal papilla. The pear-shape structure of the dermal papilla was extruded out by applying gentle pressure, and was then transferred and scratched onto a culture dish containing complete DMEM supplemented with 10% FBS and 1% of an antibiotic–antimycotic. The dermal papilla was incubated at 37˚C under a humidified atmosphere of 95% air and 5% $CO_2$ for 2 to 4 weeks, or until the dermal papilla was attached. Once the first cell migration became apparent, the medium was thereafter changed twice per week [32]. Once the cells reached full confluency, they were sub-cultured by using 0.25% trypsin/EDTA. The DPCs in their third to fifth passages were used in our experiments.

## Detection of the 5α-R and of the androgen receptor in primary DPC cultures

RT-PCR was used in order to check for the expression of target genes in primary DPCs, including those of 5α-R1, 5α-R2, and AR. The total RNA from DPCs ($1 \times 10^5$ cells/well in 6-well plates) were extracted by using the GENEzol™ reagent according to the manufacturer's instruction. cDNA was synthesized by using the revert-aid premium reverse transcriptase. The forward and reverse primers used for gene amplification are listed in Table 1. The PCR products were analyzed by using 1% agarose gel electrophoresis.

## Cytotoxicity assay

The optimal non-toxic concentrations of the AM extract and of the AC used in this study were determined by using the PrestoBlue® reagent (Invitrogen, USA). DPCs at a density of $5 \times 10^3$ cells/well in 96-well plates, were cultured overnight in complete DMEM. The cells were separately treated with various concentrations of the AM extract (0–40 μg/mL) and of the AC (0–40 μM). Cell viability was measured after an incubation with PrestoBlue® in Roswell Park

**Table 1. Forward and reverse primers, and the expected sizes of the 5α-R enzymes, the AR, and β-actin.**

| Name | Primer pair | Expected size (bp) |
|---|---|---|
| **5α-R type 1**<br>**Genbank: NM_001047.2** | F: 5′ CTGATGCGAGGAGGAAAGCCTATGC3′<br>R: 5′ GTCCAGATGCCTTTGCCTCACCTTG 3 | 1,025 |
| **5α-R type 2**<br>**Genbank: NM_000348.3** | F: 5′ TGCCTTCTGCACTGGAAATGGAGTC3′<br>R: 5′ GGAGTGGGTTTGCTCTGGGTCTTTG3′ | 801 |
| **AR**<br>**Genbank: NM_000044.3** | F: 5′ CGTGCGCGAAGTGATCCAGAA 3′<br>R: 5′ TGCGCTGTCGTCTAGCAGAGAA 3′ | 811 |
| **β-actin (internal control)**<br>**Genbank: NM_001101.3** | F: 5′ ATGATGATATCGCCGCGCTC 3′<br>R: 5′ GCGCTCGGTGAGGATCTTCA 3′ | 584 |

Memorial Institute (RPMI) medium (Invitrogen, USA) at 37˚C, for 48 h. If there were viable cells, PrestoBlue® would change from a blue to a purple-pink color, that could be spectrophotometrically detected at 570 nm.

## Determination of the 5α-R inhibitory activity

The 5α-R inhibitory activity of AM and AC was determined by using a combination of a cell (DPC)-based assay and high-performance TLC detection, as described previously [28]. Briefly, the DPCs at a density of $2 \times 10^5$ cells/mL were seeded into 6-well plates, and were treated separately with 1 mL of 30 μM T and 1 mL of 0.1% DMSO (internal control), 1 mL of 30 μM T and 1 mL of 10 μg/mL of AM, 1 mL of 30 μM T and 1 mL of 10 μM of AC, and 2 mL of 0.1% DMSO for the negative control. After a 36-h incubation, the culture medium was collected and the attached cells were tested for cell viability by using 1× PrestoBlue® reagent in RPMI medium. T and its product, DHT, were extracted from the cell culture medium by using an equal volume of ethyl acetate. The ethyl acetate fraction was dried and reconstituted with 20 μL of methanol, prior to spotting on a TLC silica gel 60 F254 aluminum plate. The TLC plate was developed by using toluene, acetone, and acetic acid at a ratio of 8:2:0.2 as the mobile phase. The developed TLC plate was then dipped in a solution of 42.5% phosphoric acid, and was heated at 120˚C for 20 min, thereby allowing for the visual detection of DHT at 366 nm by using a TLC Reprostar imager. The amount of DHT was quantified by using an image analyzing program. The inhibitory activity was determined through the decrease in the DHT production observed relatively to the internal control.

## Cell cycle analysis

DPCs at density of $2 \times 10^5$ cells/mL were seeded into 6-well plates, were further cultivated for 24 h, and were treated separately with the AM extract (10 μg/mL) and the AC (10 μM) for 48 h. After 48 h of incubation, the cells were harvested and centrifuged for 5 min at 1,200 rpm. Subsequently, the supernatant was removed, and the cells were resuspended in 0.5 mL of PBS. We added 4.5 mL of 70% ethanol in tubes, and kept them on ice. Subsequently, 0.5 mL of the cell suspension was transferred into the tubes containing the cold 70% ethanol fixative, and the cells were kept in the fixative for time ≥2 h, on ice. The ethanol-suspended cells were then centrifuged for 5 min at 1,200 rpm, in thoroughly decanted ethanol. The cells were washed three times with PBS by resuspending the pellet in 2 mL of PBS, and by centrifuging again in order to remove the supernatant. The cells were resuspended in PBS containing RNase A at 100 μg/mL, and were incubated at 37˚C for 30 min. A propidium iodide (PI) solution was added to a final concentration of 50 μg/mL, and the mixture was then incubated at room temperature for 30 min (while protected from light). Samples were analyzed for their DNA content by a FAC-Sort flow cytometer (Becton Dickinson, Rutherford, NJ, USA), with the excitation set at 488

nm and the emission filter set at 600 nm. The CellQuestTM Pro software (Becton Dickinson) was used for the identification of the sub G0/G1 phase DNA distribution in the apoptosis process. Ten thousand cells in each sample were analyzed and expressed as a percentage of total cells.

## Determination of the gene expression of hair-growth factors

The expression of four targeted hair-growth factors (namely, IGF-1, HGF, VEGF, and KGF) was determined by RT-PCR (transcriptional level). The primers were designed by using the NCBI primer design tool from the full-length mRNA sequences obtained from the NCBI GenBank, and were made to order by 1$^{st}$ Base Laboratories (Selangor, Malaysia). The forward and reverse primers for amplification are listed in Table 2. DPCs at a density of $2 \times 10^5$ cells/mL were seeded in a culture dish for 24 h, followed by the addition of various concentrations of the AM extract and of the AC, and an incubation for 24 and 48 h. The total RNA from each treatment was then extracted from the cells by using the GENEzolTM reagent. The PCR products were run on a 1% agarose gel electrophoresis, were visualized by ethidium bromide staining, were de-stained with water, and were visualized under a UV trans-illuminator by using a gel documentation system.

## Effect of the AM extract on the protein expression of hair-growth factors

DPCs at a density of $2 \times 10^5$ cells/well were seeded into 6-well plates for 24 h, and were cultured in the presence of the AM extract (10 μg/mL) for 48 h. After washing with PBS, the lysates of the treated cells were prepared by incubating the cells for 30 min in ice-cold lysis buffer containing 20 mM Tris·HCl (pH 7.5), 0.5% Triton X, 10% glycerol, 150 mM sodium chloride, 50 mM sodium fluoride, 1 mM sodium orthovanadate, 50 mM sodium fluoride, 100 mM phenylmethylsulfonyl fluoride, and a commercially available protease inhibitor cocktail (Roche Molecular Biochemicals). The cell lysates were centrifuged at 12,000 rpm, at 4°C, for 15 min, and the supernatant was collected and determined for its total protein content by the BCA protein assay kit (Bio-Rad, Hercules, CA). An equal amount of 50 μg of protein from each sample was boiled in Laemmli loading buffer at 95°C, for 5 min, so as to induce protein denaturation. The proteins were subsequently loaded on 10% SDS-polyacrylamide electrophoresis gels. After separation, the proteins were transferred onto 0.45 μm nitrocellulose membranes (Bio-Rad, Hercules, CA). Following a blocking with 5% nonfat milk in TBST (25 mM Tris·HCl (pH 7.5), 0.05% Tween-20, and 125 mM NaCl) for 2 h, the membranes were incubated with the appropriate primary antibodies against Bcl-2, Bax, caspase-3, VEGF, KGF, HGF, and BMP-4, for 10 h, at 4°C. The resulting membranes were then washed three times

**Table 2. Forward and reverse primers and the expected sizes of growth factors and β-actin.**

| Name | Primer pair | Expected size (bp) |
|---|---|---|
| **IGF-1**<br>Genbank: NM_000618.3 | F: 5′ CTCCTCGCATCTCTTCTACC 3′<br>R: 5′ GTTTCCTGCACTCCCTCTAC 3′ | 361 |
| **FGF-7**<br>Genbank: NM_002009.3 | F: 5′ ATGAACACCCGGAGCACTAC 3′<br>R: 5′AAATCTCCCTGCTGGAACTG 3′ | 653 |
| **HGF**<br>Genbank: NM_000601.4 | F: 5′CAGAGGTACGCTACGAAGTC3 ′<br>R: 5′GATGTGCCACTCGTAATAGG 3′ | 1,320 |
| **VEGF**<br>Genbank: AAA35789.1 | F: 5′ACCCATGGCAGAAGGAGGAG 3′<br>R: 5′CCTTGCAACGCGAGTCTGTG 3′ | 440 |
| **β-actin (internal control)**<br>Genbank: NM_001101.3 | F: 5′ATGATGATATCGCCGCGCTC 3′<br>R: 5′ GCGCTCGGTGAGGATCTTCA 3′ | 584 |

with TBST (for 15 min), and were incubated with horseradish peroxidase-coupled secondary antibodies for 1 h, at room temperature. The signals from the target proteins were detected by the use of a chemiluminescence substrate (Super Signal West Pico, Pierce, Rockford, IL). The intensity of the protein signal was quantified by using unsaturated images on the ImageJ 1.53e software.

## Determination of the inhibitory effect on the DHT-AR complex formation and translocation from the cytoplasm to the nucleus

In this study, the immunofluorescence staining of the AR was used in order to determine the possible effect of the AM extract and of the AC on the DHT-AR complex formation and translocation from the cytoplasm to the nucleus of DPCs. This was performed by using the method previous described [33]. Practically, DPCs were first seeded into 24 wells-plate, and were cultured for cell attachment for 24 h. The cells were then treated separately with the AM extract (10 μg/mL) or the AC (10 μM) in the presence of 30 μM of DHT. The positive control was incubated with 30 μM of DHT and 20 nM of flutamide (an androgen-AR binding inhibitor). After the treatment, the cells were washed three times in ice-cold PBS, and were fixed with absolute methanol for 15 min at 20˚C. After fixation, the cells were washed three times in ice-cold PBS, and were incubated with 2% (v/v) of BSA in PBS-T (PBS with 0.5% TritonX-100; nonionic detergent) for 1 h in order to permealize and block the cells, so as to allow the antibody to access intra-organelle antigens and to block nonspecific antiserum binding. This step was followed by an overnight incubation (at 4˚C with a monoclonal rabbit anti-AR antibody diluted in blocking buffer. After washing off the primary antibody with PBS-T, the cells were incubated (for 4 h in the dark) with a secondary Alexa Flur488-labeled donkey anti-rabbit IgG antibody (1:600; Molecular Probes) diluted in blocking buffer. After removing the secondary antibody, the cells were washed three times with PBS-T, and their nuclei were counterstained with 1 μg/mL of Hoechst for 15 min. The cells were imaged by using fluorescence spectroscopy at room temperature.

## Statistical analysis

All data were obtained from three individual experiments. The obtained data are presented as mean ± SD, and were analyzed by using one-way analysis of variance (ANOVA) via GraphPad Prism 9.0. Statistical significance was considered for p values lower than 0.05.

## Results and discussion

### Isolation of DPCs and expression of target genes

DPCs were isolated by a simple micro-dissection from the bulb of the dissected hair follicles obtained from the occipital scalp region of a patient undergoing hair transplant. The DPCs with membrane-bound, pear-shaped structures were isolated from an intact hair follicle (Fig 1Aa and 1Ab). Structurally, the DPCs are enveloped with membrane-bound, pear-shaped structures; we, thus, needed to scratch the membrane part of the DPCs in order to allow for the whole pear-shaped structure to come out and stay at the bottom of the culture dish, and then open the explant and leave it untouched until the migration of the initiated cells from the explant; a migration that was observed after 14–30 days [34]. Fig 1Ac–1Ae) show the migration of the cells from the explant in the form of a spread-out growth, until they reach full confluency. As previously reported [33–40], the isolated DPCs appeared to have specific characteristics, including a flattened morphology, an irregular spindle shape, and an ability to form multi-layered aggregations at full cell confluency (Fig 1Af–1Ah).

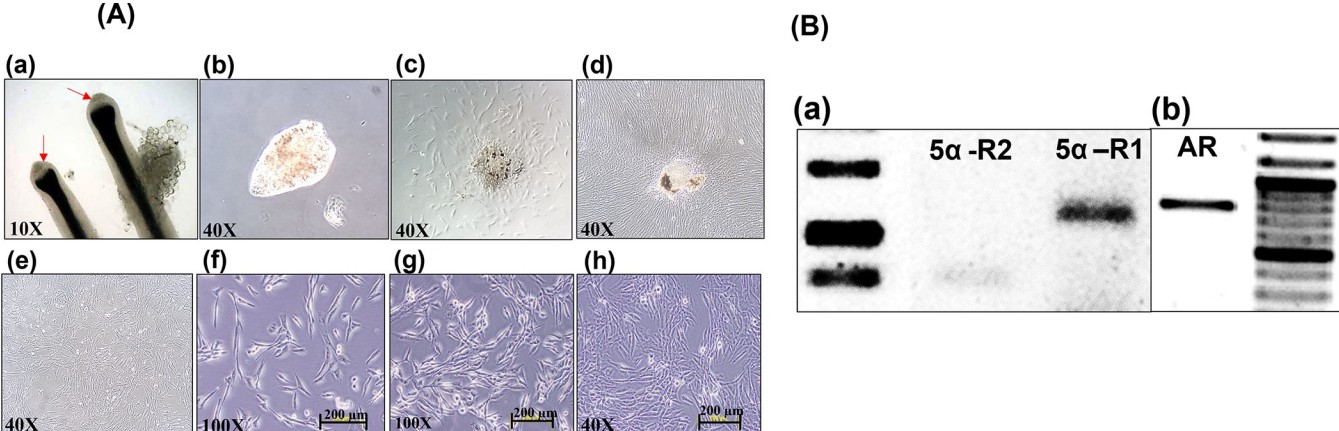

**Fig 1. The culture characteristics of DPCs isolated from human hair follicles obtained by AGA patients.** (A) (a) Intact hair follicle bulb (red arrows indicate the dermal papilla in the hair bulb region; 10×). (b) The pear-like shape of the typical membrane-bound DPCs (40×). (c) The initial outgrowth of isolated DPCs after 2 weeks of culturing (40×). (d) The isolated DPCs before full cell confluency after 1 month of culturing (40×). (e) The isolated DPCs at full cell confluency (40×). (f and g) The isolated DPCs at low cell density after culturing for 1 and 3 days, exhibiting a flatted and spindle-shaped morphology (100×). (h) The isolated DPCs forming multi-layered aggregations before reaching full confluency (40×). (B) RT-PCR showing the expression of target genes by DPCs. (a) The expression of 5α-R1 (1,025 bp) and 5α-R2 (801 bp) at passage 7. (b) The expression of AR (801 bp) at passage 5.

Since the action of androgens causing AGA is known to be exerted through the functions of 5α-R and the AR, the gene expression of the two targets in the DPCs was a prerequisite for studying the mechanistic effects of AM and AC as anti-AGA agents. These gene expressions were determined by RT-PCR by using specific primers for the two 5α-R isoforms (5α-R1 and 5α-R2) and the AR protein. The results revealed that both the AR and 5α-R1 were strongly expressed in primary DPCs, along with a much lower expression of the 5α-R2 (Fig 1B). Similar results have been previously reported regarding the AR and 5α-R1 mRNAs expressed in all parts of the hair follicle (including the DPCs), while the 5α-R2 mRNA has been previously shown to be expressed only in the mesenchymal parts of the dermal papilla and in the connective tissue sheath [41]. The same report has also shown that the intensity of the expression of these genes in each part of the hair follicle did not differ between follicles from balding and those from nonbalding scalps [41]. As far as the AR expression is concerned, it has been demonstrated by hormone binding assays and RT-PCR that the AR expression is significantly higher in bald DPCs than in nonbald ones [42]. Thus, the strong expression of 5α-R1 (Fig 1Ba) and AR (Fig 1Bb) in our primary DPCs can be used as a model for investigating the mechanism of action of the AM extract and of AC.

## The effect of the AM extract and of AC on the cell cycle of isolated primary DPCs

Hair follicle morphogenesis is characterized by a tightly regulated balance of cell proliferation, differentiation, and apoptosis [43]. In order to observe the effects of AM and AC on the cell cycle of isolated DPCs, the primary cells were first determined for the cytotoxicity of both the extract and active compound by using MTT assay for 48 hr. This incubation time was optimized based on the time-course study of 5α-R enzyme activity assay (0, 6, 12, 24, 36 and 48 h) using the primary DPCs (S6 Fig in S1 File). The results showed that the treatment of AM at the concentration range of 2.5–20 μg/mL and AC at 2.5–20 μM did not cause significant cytotoxicity on DPCs, compared with the non-treated cells (Fig 2). At the maximum non-toxic concentrations of 20 μg/mL AM and 20 μM AC, the cell viability was observed to be in the range of 80–85% which is lower than the acceptable 90% cell viability. Therefore, we selected the

**(a)**

**(b)**

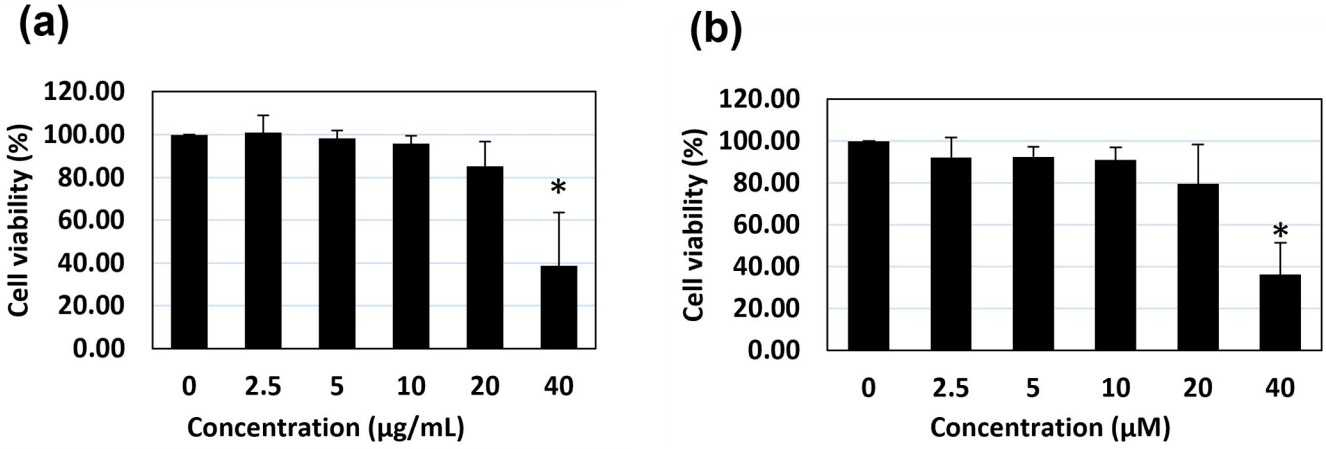

**Fig 2. Cytotoxicity assay of AM and AC on DPCs.** DPCs were treated with various concentrations of (a) AM (0–40 μg/mL) and (b) AC (0–40 μM) for 48 h. Cell viability was evaluated by PrestoBlue® assay. The data is presented as mean + SD (n = 3). * $p < 0.05$ versus non-treated cells.

concentrations of 10 μg/mL AM and 10 μM AC for cell cycle analysis and further experiments. DPCs were treated with AM (10 μg/mL) and AC (10 μM) for 48 h, followed by an analysis of their cell phase distribution by a flow cytometer. The results showed that both the AM extract and the AC significantly decreased the number of cells in the G0 phase, and simultaneously increased those in the G2-phase (as compared to the control; Fig 3). On the other hand, no significant differences were observed between the treated and the non-treated cells in terms of the cell population numbers undergoing their G1 and S phases. Since the G0 cells represent apoptotic cells and the G2 phase represents the active phases of DNA synthesis and cell

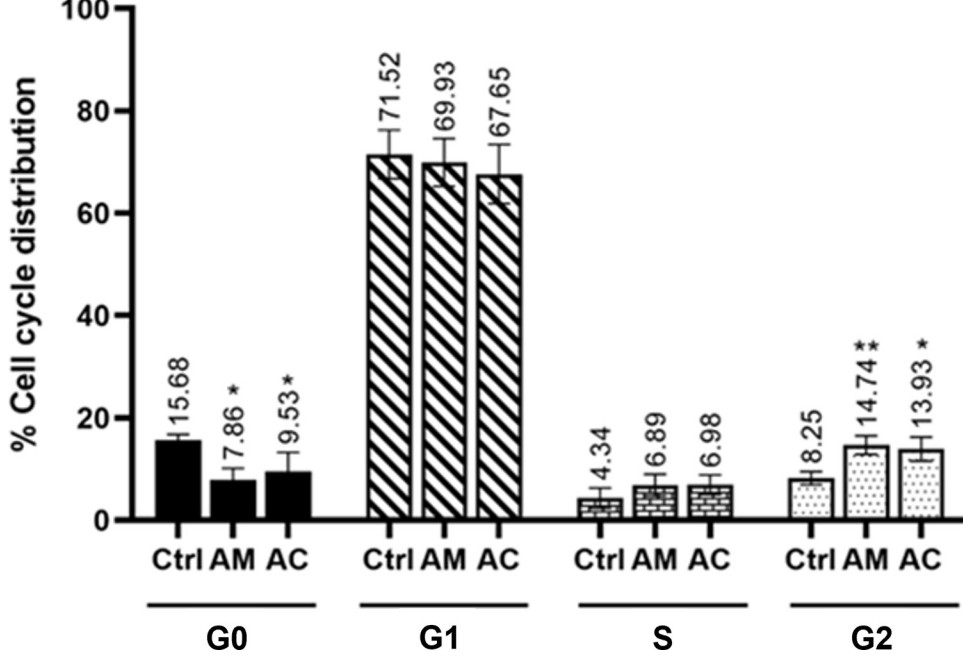

**Fig 3. Cell cycle analysis of DPCs after a treatment with the AM extract or with AC.** The cells were treated with either the AM extract (10 μg/mL) or AC (10 μM). After 48 h, the cells were analyzed for DNA content by a FACSort flow cytometer, and their sub G0/G1 phase population was evaluated for its DNA distribution in the apoptotic process. The data are presented as mean ± standard deviation (SD) (n = 3). *$p<0.05$ and **$p<0.01$ versus non-treated cells.

proliferation, the obtained results suggest that AM and AC not only prevent the apoptosis of DPCs, but also induce the proliferation of these cells.

## The effect of the AM extract and of AC on the inhibition of 5α-R

By using the DPC-based assay coupled with a non-radioactive thin layer chromatography (TLC) for determining the 5α-R activity [28], both the AM extract and the AC were found to clearly inhibit 5α-R (Fig 4), Quantitatively, the AM extract at 10 µg/mL exhibited a 73.40% (±8.40%) inhibition (lane 3), while AC at 10 µM (or 2.56 µg/mL) exhibited a 67.71% (±3.81%) inhibition (lane 4). Dutasteride (DU; a competitive inhibitor of 5α-R used as a positive control at 1 µM) exerted a 100% inhibition of 5α-R (lane 2). The obtained results suggest that both the AM extract and the AC could decrease the 5α-R activity, and cause a reduction of the DHT production by more than 50% in DPCs. Moreover, the AM extract (at 10 µg/mL) exhibited the strongest 5α-R inhibitory activity compared to other potential extracts and pure compounds (S2 Fig in S1 File); notably, with a similar level of potency to the commercial inhibitor of finasteride (1µM). This may reflect the synergistic effects of the constituents present in the AM extract [29]. To prove this, however, it is necessary to find other active constituents in AM, followed by observing the 5α-R inhibitory activity of their combinations with AC.

## The effect of the AM extract and of AC on the inhibition of the DHT-AR complex formation and of its nuclear translocation

AGA is an androgen-mediated disorder the mechanism of which involves the nuclear translocation of the AR upon the binding of androgens, thereby resulting in the activation their

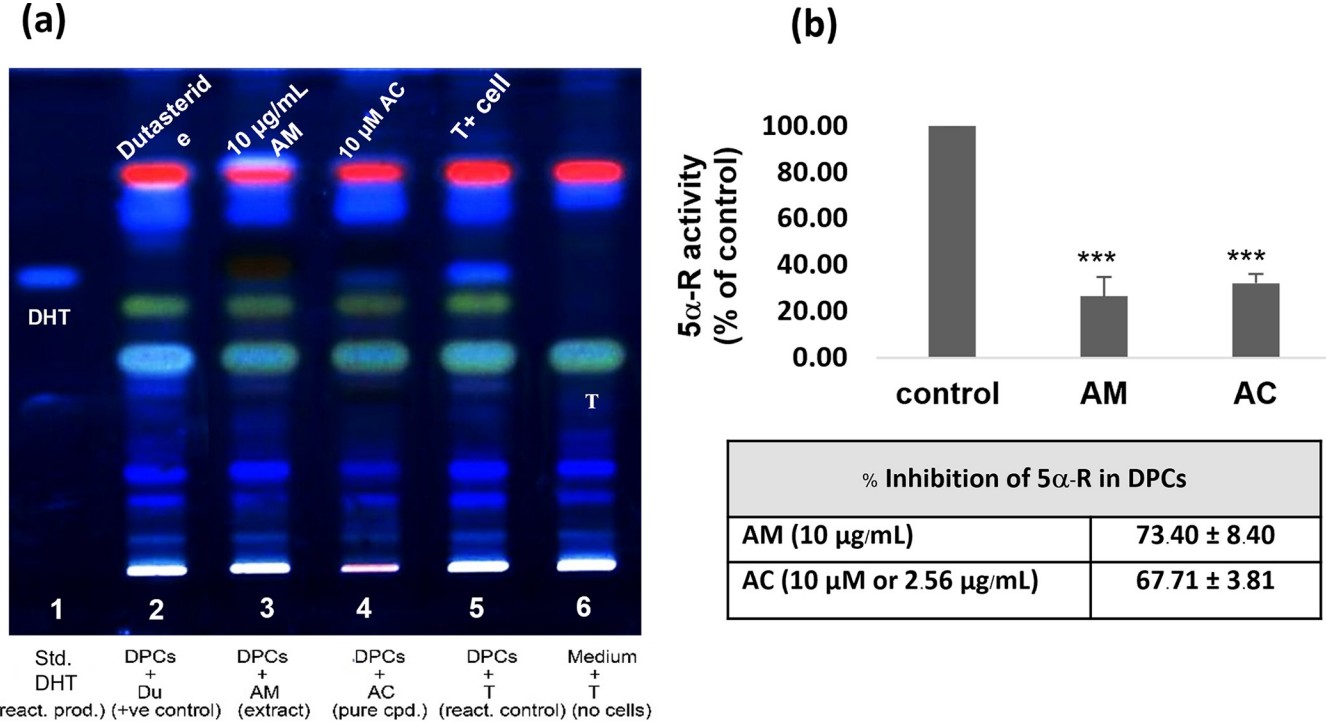

**Fig 4.** 5α-R inhibitory effects of AM and AC by performing on TLC plate (a) A TLC plate showing the 5α-R inhibitory activity of the AM extract (10 µg/mL) (lane 3) and of AC (10 µM) (lane 4), with dustasteride (1 µM) being used as a positive control (lane 2). The internal control (cells without the inhibitor) is shown in lane 5, while the controls with standard DHT and cells with T are shown in lanes 1 and 6, respectively. (b) The bar graphs represent the percentage of DHT produced in the presence of the AM extract and the AC relatively to the internal control, with the obtained values presented in the Table. All data were obtained from experiments performed in triplicate, and are presented as mean ± SD (n = 3). ***p<0.005 versus non-treated cells.

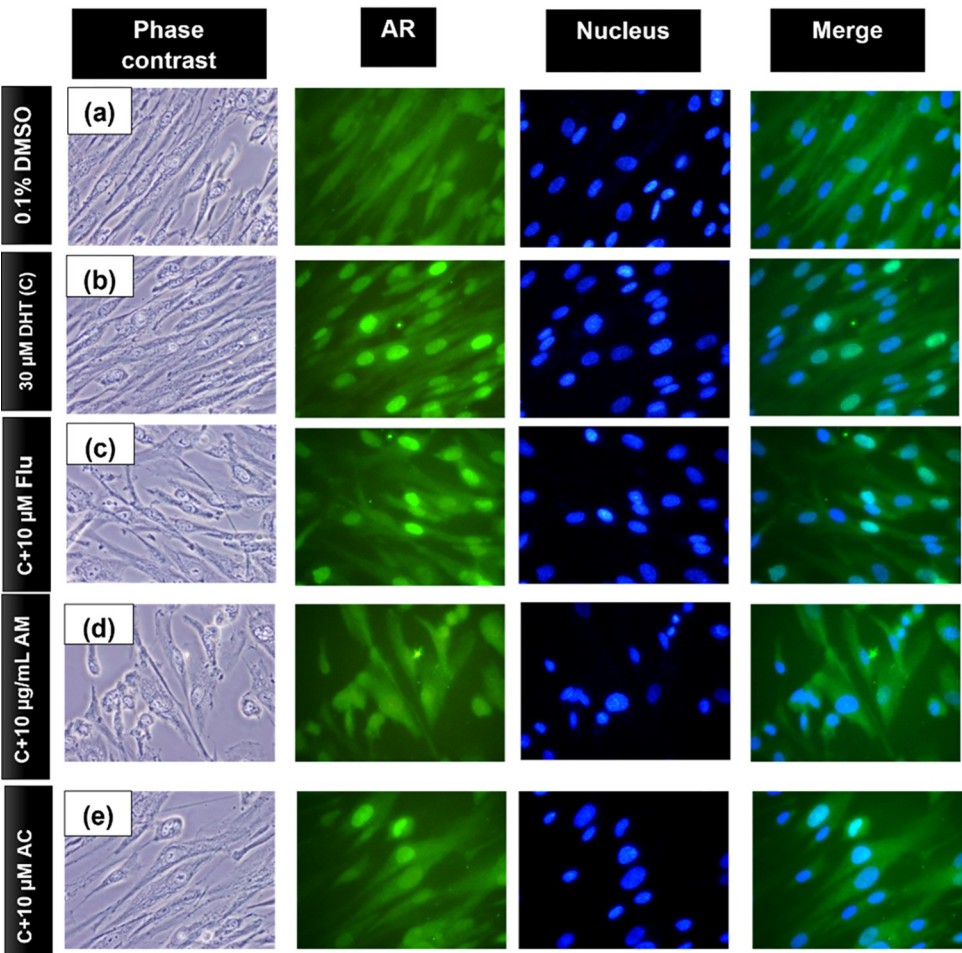

**Fig 5. The immunofluorescent staining of the AR and the nucleus.** The first column shows phase contrast pictures of DPCs, the second column shows the pictures of the AR green fluorescence staining by using an antibody against AR, the third column shows the blue fluorescence nuclear staining, and the fourth column shows the merged images of the second and third columns. Rows: (a) DPCs under normal conditions (without a DHT treatment); (b) DPCs treated with 30 μM of DHT for 12 h; (c) DPCs treated with 30 μM of DHT and 10 μM of flutamide (positive control); (d) DPCs treated with 30 μM of DHT and 10 μg/mL of the AM extract; (e) DPCs treated with 30 μM of DHT and 10 μM of AC. The nucleus was stained by Hoechst dye blue fluorescence and observed under a fluorescence microscope (40×).

androgenic activity. Therefore, the effects of the AM extract and of AC on the nuclear translocation of AR in DPCs were investigated. In this study, DHT (at 30 μM) was used as the active androgen for stimulating its binding with AR and for causing the nuclear translocation of the DHT-AR complex. Flutamide was used as positive control in this study. Flutamide is a selective antagonist of the AR, where it competes with the androgens T and DHT for its binding to the AR in various target tissues [35]. The results of the nuclear translocation were obtained by using the immunofluorescent staining of AR and the blue fluorescence (Hoechst dye) staining of nuclei. As shown in Fig 5, it was found that DHT readily activated the translocation of the AR (presumably in the form of a DHT-AR complex) from the cytoplasm into the nucleus, as observed by the much higher intensity of the green fluorescence in the nucleus than in the cytoplasm (Fig 5B) when compared with the control (Fig 5A). In the presence of the active AM extract (10 μg/mL) (Fig 5D) or the AC (10 μM) (Fig 5E), it appeared that both preparations could counteract the effect of DHT by exhibiting a marked decrease in the levels of nuclear

translocation of the AR (Fig 5D and 5E). Interestingly, the efficiency of the positive control (containing 10 μM flutamide) in inhibiting the nuclear translocation of the AR (Fig 5C) was found to be lower than those of the AM extract and the AC. Furthermore, the AM extract and the AC exhibited a reduction of the AR translocation after a stimulation with T at 30 μM (S3 Fig in S1 File). These results suggested that the AM extract and the AC were able to inhibit the both DHT-AR and T-AR complex formation and translocation from the cytoplasm into the nucleus (which is the site of the DHT action) in DPCs. Whether the inhibition took place at the step of the binding of the DHT to the AR or at the complex translocation step remains unclear. However, our finding is in agreement with those of previous reports showing that the extract of *Scutellaria baicalensis* and its active compound (baicalin) were capable of inhibiting the translocation of the AR, with implications for the prevention of AGA [33, 36].

## The effect of the AM extract and of AC on the gene expression of hair growth factors

As mentioned earlier, the overproduction of DHT in AGA patients alters the production of some growth factors that are secreted by DPCs (particularly, IGF-1, KGF, HGF, and VEGF); these growth factors are known to be involved in the positive regulation of the hair growth cycle [6–8]. In an attempt to examine whether the AM extract and the AC have any effect on the mRNA expression of these growth factors, the RT-PCR technique was used in order to quantitate their gene expression. It was found that both the AM extract and the AC could significantly increase the mRNA levels of VEGF and HGF in the presence of T (30 μM) (Fig 6A and 6B), without affecting those of KGF and IGF-1 (S4 Fig in S1 File). The upregulation of VEGF was found to be highest at 6 h, with approximately a 3.5-fold increase (as compared to the control; Fig 6A), followed by its continuous decrease to nearly the control level at 48 h. In the case of HGF, on the other hand, its gene expression in response to a treatment with the AM extract and the AC appeared to be slower, with a 2.2- and a 4.8-fold increase of the HGF mRNA at 24 and 48 h, respectively (Fig 6B). Therefore, it seems that the induction of the gene expression of VEGF (6 h) in response to a treatment with either the AM extract or the AC, is

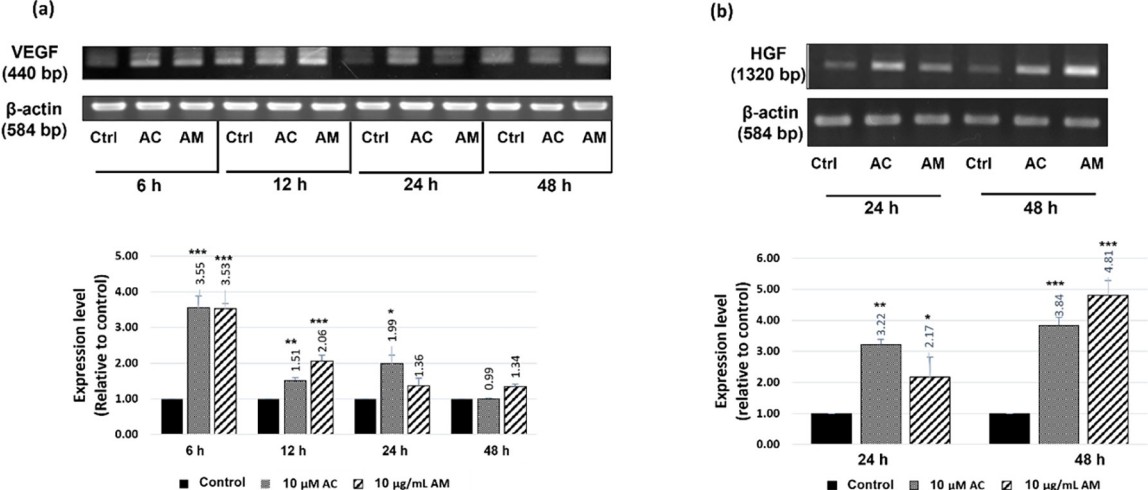

**Fig 6. The effect of the AM extract (10 μg/mL) and of the AC (10 μM) on the gene expression of VEGF and HGF.** DPCs were incubated with 30 μM of T, and were treated with the AM extract or with AC at various times before determining the gene expression of (a) VEGF and (b) HGF. The expression levels of each of these growth factors were analyzed by using the ImageJ 1.53e software. All data were performed in triplicate, and are herein presented as mean ± SD (n = 3). *p<0.05, **p<0.01, and ***p<0.005 versus non-treated cells.

quicker than that of HGF (24 h). In the literature, the expression of VEGF in DPCs during the hair cycle has been reported to fluctuate, with a strong expression being recorded during the anagen phase, and lower expressions being recorded during the catagen and the telogen phases [18]. Similarly, the upregulation of the HGF mRNA has also been observed in the anagen phase during the DNA synthesis in keratinocytes deriving from the human hair follicle development [37]. Our results are similar to those of a previous report showing that the Geranium sibiricum extract promoted hair growth (both *in vitro* and *in vivo*) by upregulating the mRNA levelS of VEGF and HGF, and that the reduction in the expression of these genes led the cells to the catagen phase [37]. For KGF and IGF-1, the observed negative induction in their mRNA expression suggested that neither of these hair growth factors are targets of the AM extract and of the AC. Alternatively, a reason for our observations might be that the time of their response was not covered by herein adopted timeframe of study.

### Effect of the AM extract on the protein expression of hair growth factors

In addition to the effect on the mRNA expression of hair growth factors, the effect on the protein expression of these genes was also examined. This was carried out by Western blotting with a focus on the hair growth factors KGF, HGF, and VEGF. In this experiment, only the AM extract was used for the cell treatment, as all results obtained above revealed similar effects between both the AM extract and the AC. The concentration of the AM extract was, again, set at 10 μg/mL; a concentration that has been shown to be non-toxic (Fig 2). Our findings revealed that the AM extract was able to significantly increase the protein expression of all three growth factors (Fig 7). In fact, it can be seen that the AM extract can significantly increase the protein levels of HGF after 12 and 24 h of incubation (Fig 7A), and those of KGF after 6, 12, and 24 h of incubation (Fig 7B). As far as the VEGF is concerned, its protein overexpression was found to increase significantly at 6 h, followed by a rapid decline at 12 h (Fig 7C). By comparing between the results of the mRNA expression (Fig 6) and those of the protein expression (Fig 7), both HGF and VEGF seem to exhibit a good correlation between the two expression levels in terms of their time responses to the AM extract treatment. KGF, on the other hand, was shown to be well responsive to the AM extract treatment only at the protein expression level (Fig 7B), but not at the mRNA expression level (S3a Fig in S1 File). It might be possible that the endogenous mRNA of KGF in DPCs was already available as a result of normal gene expression and, thus, the cells were able to readily respond to the AM extract treatment through the protein expression of KGF. Therefore, all results suggest that the AM extract can stimulate the production of hair growth factors such as the VEGF, the HGF, and the KGF.

### Effect of the AM extract on the protein expression of anti-apoptotic mediators

Apoptosis plays a central role in the regulation of the hair follicle during the catagen phase [38]. Its effects are modulated by a complex signaling pathway involving the important controlling mechanisms of Bcl-2, Bax, and caspase-3 [38]. In an attempt to evaluate whether the AM extract (10 μg/mL) affects the hair-inducing activity of DPCs, the expression levels of these associated proteins were examined. Our results revealed that the AM extract was capable of inducing the protein expression of Bcl-2 by a significant 75% after 6 h of incubation (Fig 8A). Simultaneously, the protein levels of Bax and caspase-3 were found to be considerably decreased after 6–24 h of incubation (Fig 8B and 8C). These results are similar to those obtained after a minoxidil treatment [18] or a treatment of DPCs with macrophage-derived extracellular vesicles [17], that promoted the migration and the proliferation of DPCs as well

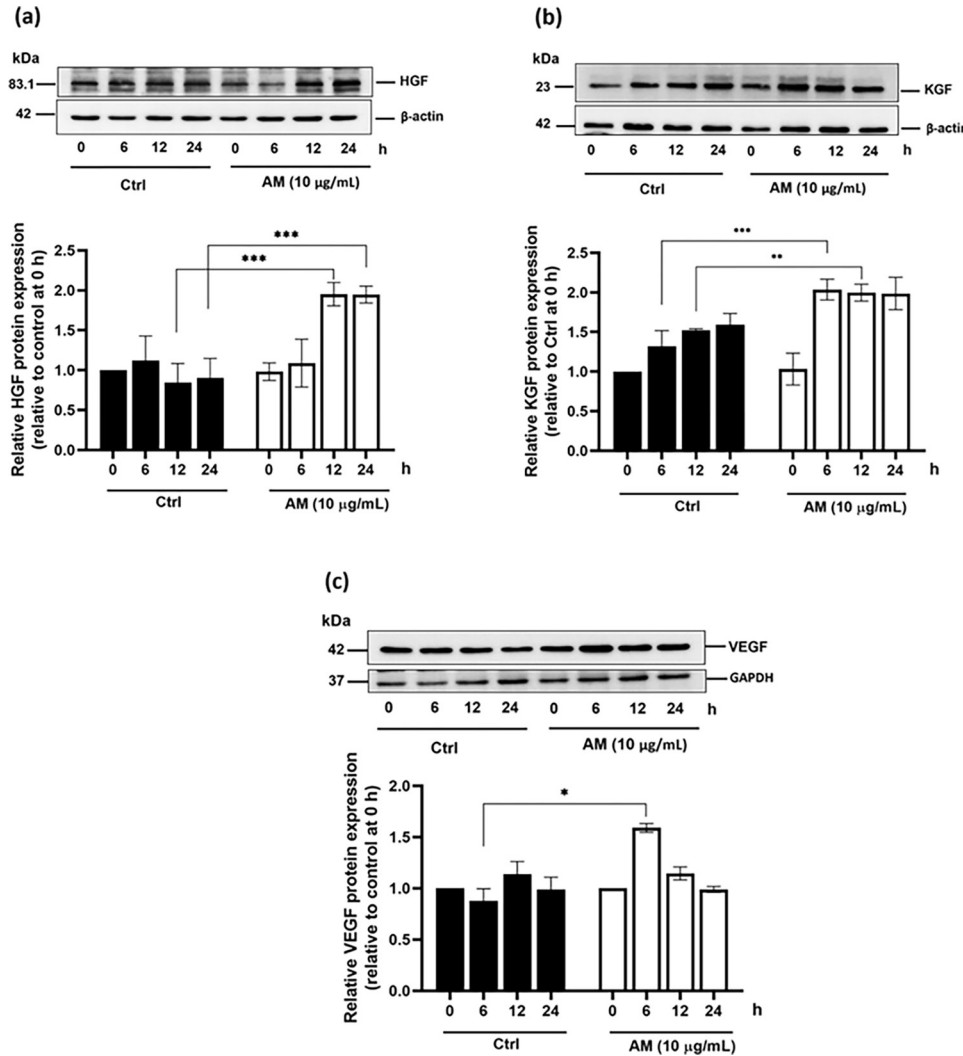

**Fig 7. The effect of the AM extract on the protein expression of the hair growth factors.** (a) VEGF, (b) HGF and (c) KGF. DPCs were incubated with T (30 μM), and were treated with the AM extract (10 μg/mL) for various periods (6, 12, and 24 h). DPCs incubated with only T (30 μM) acted as a control. The protein expression is represented relatively to the control group levels at 0 h. The bar graph data analysis of the undertaken Western blotting shows the expression of VEGF (a), HGF (b), and KGF (c). All data are presented as mean ± SD (n = 3). *$p < 0.05$, **$p < 0.01$, and ***$p < 0.005$ versus non-treated cells.

as the delay of the apoptotic process. Finasteride, a specific 5α-R type 2 inhibitor, has been found to affect the caspase expression in the hair follicles, thereby causing an induction of the active growth hair cycle [39]. Tetrahydroxystilbene glucoside isolated from Polygonum multiflorum has also been reported to possess an anti-hair loss activity, by suppressing Bax that induces apoptosis [40]. Therefore, our findings support the ability of the AM extract to promote hair growth by delaying the initiation of the catagen and/or the apoptotic phase in the hair cycle.

## Effect of the AM extract on the protein expression of BMP-4

The BMP pathway is a key regulator of the genetic program controlling the hair shaft differentiation in the hair follicle. It has been reported that a negative effect of the BMP signaling can

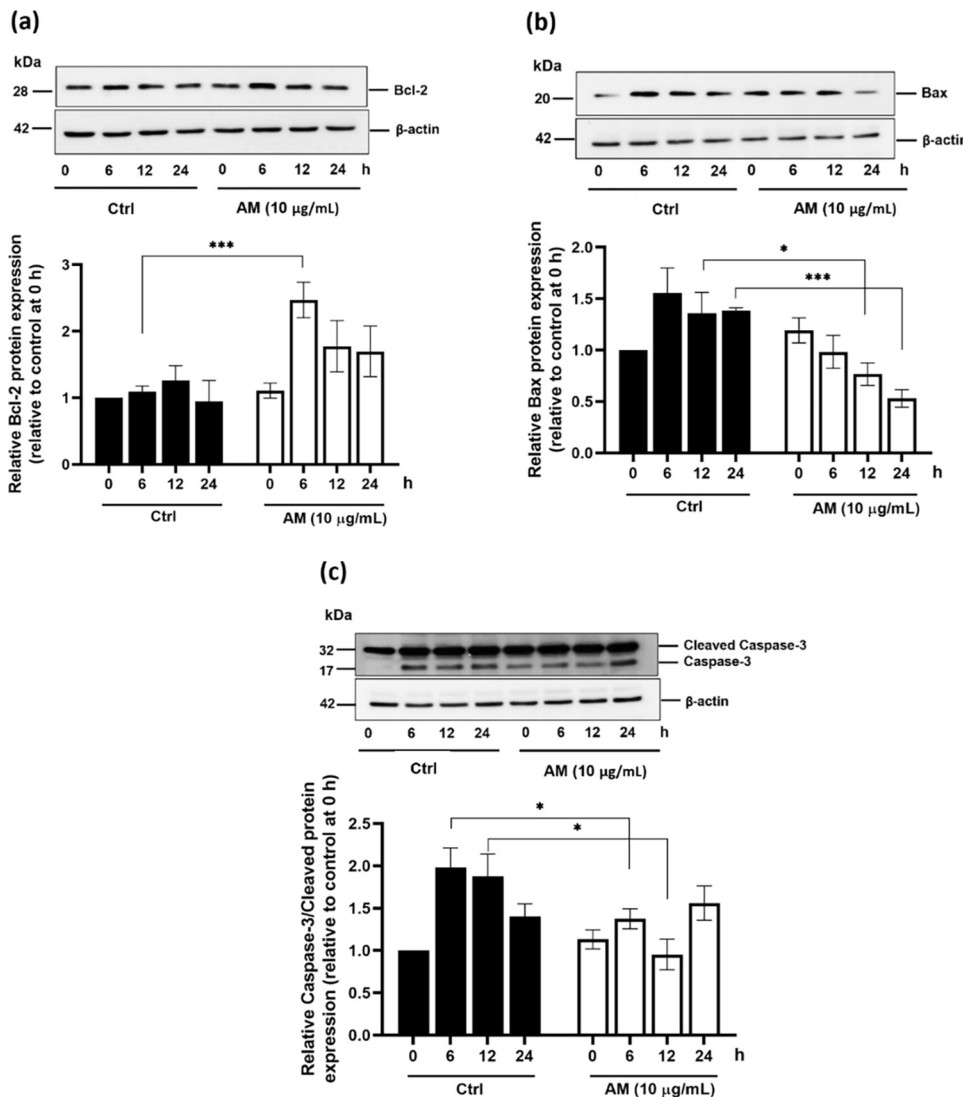

**Fig 8. The effect of the AM extract on the protein expression of anti-apoptotic mediators.** DPCs were incubated with T (30 μM) as a control, and were treated with the AM extract (10 μg/mL) for various periods (6, 12, and 24 h). The bar graph data analysis of the undertaken Western blotting is presented relatively to the control group at 0 h. The expression levels of Bcl-2 (a), Bax (b), and cleaved caspase-3 (c) are presented. All data are presented as mean ± SD (n = 3). *p<0.05, and ***p<0.005 versus non-treated cells.

induce hair growth in the hair follicle that is in its anagen phase, whereas a positive effect can promote the hair growth shifting to the telogen phase [15]. Thus, the effect of the AM extract on the expression of BMP-4 in DPCs was determined. Our Western blotting analysis revealed that the AM extract was able to continuously decrease the protein levels of BMP-4 during the 24 h of the treatment (as compared to control; Fig 9). This finding suggests that the AM extract potentially induces hair growth at the telogen phase by attenuating the BMP-4 protein in DPCs. Similarly, previous studies have shown that the enhancement of the BMP signal activation with noggin (a BMP antagonist) can result into a significant delay of the induction of the hair follicle, and lead to progressive baldness [15].

Taken together, the AM extract and the AC clearly act as 5α-R inhibitors that can significantly reduce the formation of DHT in DPCs. The pure compound of AC at 10 μM exhibited

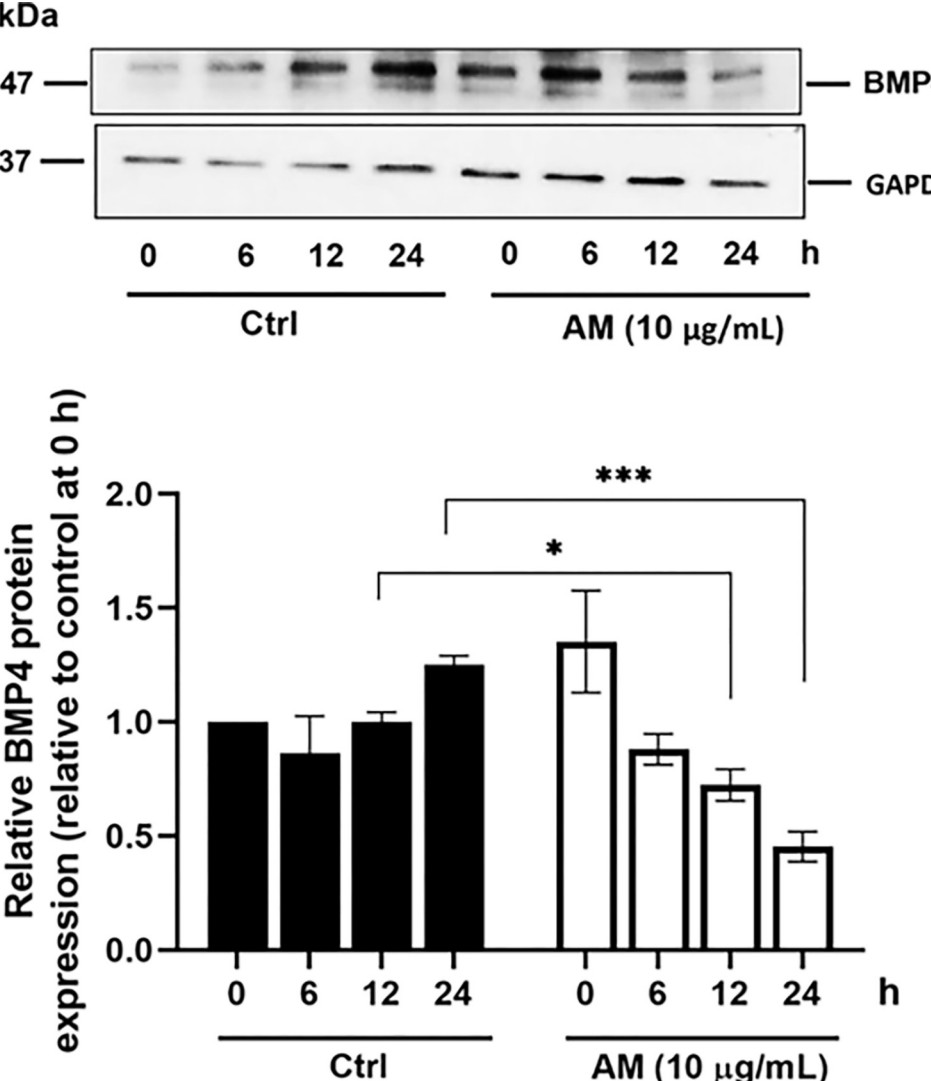

**Fig 9. Effect of AM on the protein expression of BMP-4.** DPCs were incubated with T (30 μM) and were treated with the AM extract (10 μg/mL) for 6, 12, and 24 h. DPCs were incubated only with T (30 μM) as a control. The histograms represent the relative levels of the protein expression (relative to the control group). Quantitative analysis of the protein expression was determined by using the ImageJ software. Western blot showing the protein expression of BMP-4 in each condition, and quantitative analysis of the protein expression. The data are presented as mean ± SD (n = 3). *p<0.05, and ***p<0.005 versus non-treated cells.

its highest inhibitory activity, and caused a reduction of the DHT production by 67.7% in DPCs, whereas the AM extract at 10 μg/mL was able to reduce the formation of DHT by 73.5% in DPCs. It is well-known that androgens exert their activity through binding with the AR. The reduction of DHT would, on one hand, lead to a lowering of the level of the AR-DHT complex. On the other hand, our results also suggest that both the AM extract and the AC can interfere directly with the binding of DHT to the AR, thereby resulting in a decrease of the translocation of the AR-DHT complex from the cytoplasm to the nucleus. Moreover, the treatment with the AM extract or the AC in the presence of T revealed a significant increase in the mRNA expression of the key growth factors of VEGF and HGF. These results suggest that both the AM extract and the AC can promote hair growth by stimulating the expression of

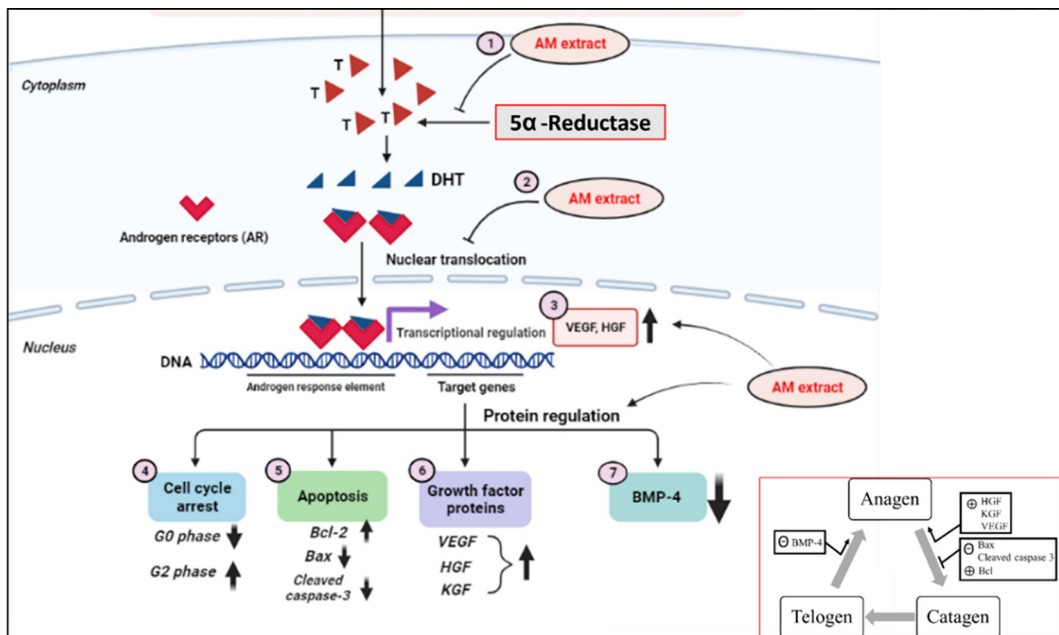

**Fig 10. A proposed mechanism of the AM extract on the modulation of the hair cycle.**

VEGF and HGF; two signal transduction molecules that regulate the hair follicle cyclic growth. Both VEGF and HGF play important roles in the angiogenesis associated with the hair growth and hair cycling [6–8]. In addition, this study also revealed that the levels of Bcl-2 (that protects hair follicle cells from apoptosis, thereby resulting in maintaining the hair follicle in the anagen-catagen phase) were significantly increased as a result of a treatment with the AM extract. On the other hand, the levels of Bax and caspases-3 were markedly decreased in DPCs, thereby resulting in an inhibition of cell death. Similarly, the AM extract strongly attenuated the protein levels of BMP-4, thereby potentially causing hair growth stimulation. These results suggest that the AM extract can promote hair growth by stimulating the expression of VEGF and HGF that proceed with an intercellular signal transduction, and can regulate the hair follicle cyclic growth (Fig 10).

The AM extract inhibits 5α-R (1) and reduces the nuclear translocation of AR (2). The AM extract upregulates the mRNA levels of VEGF and HGF (3). In the cell cycle pathway, the AM extract diminishes the cell population in the G0 phase, and increases the number of cells in the G2 phase (4). The AM extract reduces the expression of Bax and cleaved caspase-3, and induces the expression of Bcl so as to delay the induction of the catagen phase (5). The protein expression of hair growth factors (HGF, KGF, and VEGF) was found to increase after a treatment with the AM extract, thereby promoting the cells' retaining in the anagen phase (6). Moreover, the AM extract reduced the expression of BMP-4, thereby facilitating the cells' turning to the anagen phase (7).

## Conclusions

AGA is a common type of scalp hair loss in both men and women. It is a genetically determined disorder associated with the increasing production of DHT, that is formed from T by 5α-R. Androgens and their receptor (AR) cause signaling and biochemical reactions that result in the alteration of cytokines and growth factors, thereby leading to premature hair loss. Therefore, potential drugs to be used for the prevention or the cure of AGA should be able to inhibit

5α-R in order to reduce the androgen formation or to block the AR activation in DPCs. We have shown in this study that the target DPCs could be isolated by a simple microdissection without the use of enzyme treatments. The isolated DPCs clearly exhibited the expression of target genes (especially those of the 5α-R1 and the AR), thereby convincing us that these cells are a suitable model for the study of the mechanisms of action of the AM extract and of the AC. These included their effects on the 5α-R enzymatic activity, the nuclear translocation of the DHT-AR complex, and the expression of certain growth factors. Our results suggest that the AM extract as well as its active compound (AC) possess a potent in vitro 5α-R inhibitory activity coupled with an inhibition of the androgen effect in DPCs causing the decrease of the androgen-dependent activation of the AR. Furthermore, they increase the mRNA expression of some growth factors that are generally believed to stimulate hair growth. Due to our experiments being conducted on DPCs isolated from AGA patients, these finding might support the use of the AM extract and of the AC in future hair growth / protecting strategies for AGA patients. Moreover, there is a potential for the development and future application of the AM extract as a natural drug promoting hair growth, in both cosmetics and pharmaceuticals used for the treatment of AGA.

## Supporting information

**S1 File.**
(ZIP)

**S1 Raw images.**
(PDF)

## Acknowledgments

S.N. would like to thank Chulalongkorn University for providing the King Bhumibol Adulyadej's 72nd Birthday Anniversary Scholarship (72nd King's Birthday Scholarship). W.P. would like to thank The Thailand Research Fund (TRF) for her Ph.D. Scholarship through RRi Program.

## Author Contributions

**Conceptualization:** Wanchai De-Eknamkul.

**Data curation:** Woraanong Prugsakij, Sukanya Numsawat, Ponsawan Netchareonsirisuk.

**Formal analysis:** Woraanong Prugsakij, Ponsawan Netchareonsirisuk.

**Funding acquisition:** Parkpoom Tengamnuay, Wanchai De-Eknamkul.

**Investigation:** Woraanong Prugsakij, Sukanya Numsawat, Ponsawan Netchareonsirisuk.

**Methodology:** Woraanong Prugsakij, Sukanya Numsawat.

**Project administration:** Parkpoom Tengamnuay, Wanchai De-Eknamkul.

**Resources:** Wanchai De-Eknamkul.

**Supervision:** Parkpoom Tengamnuay, Wanchai De-Eknamkul.

**Validation:** Ponsawan Netchareonsirisuk.

**Writing – original draft:** Woraanong Prugsakij, Ponsawan Netchareonsirisuk, Parkpoom Tengamnuay, Wanchai De-Eknamkul.

**Writing – review & editing:** Wanchai De-Eknamkul.

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
