## [Decision Letter · Decision Letter 0]

5 Jan 2023

PONE-D-22-22814

Mechanistic synergy of hair growth promotion by the Avicennia marina extract and its active constituent (avicequinone C) in dermal papilla cells isolated from androgenic alopecia patients

PLOS ONE

Dear Dr. De-Eknamkul,

Thank you for submitting your manuscript to PLOS ONE. After careful consideration, we feel that it has merit but does not fully meet PLOS ONE’s publication criteria as it currently stands. Therefore, we invite you to submit a revised version of the manuscript that addresses the points raised during the review process.

Only a minor comments have been raised and please respond as soon as possible.

We look forward to receiving your revised manuscript.

Kind regards,

Cheorl-Ho Kim, Ph.D.

Academic Editor

PLOS ONE

https://journals.plos.org/plosone/s/fileid=ba62/PLOSOne_formatting_sample_title_authors_affiliations.pdf.

“PERDO's Center of Excellence on Medical Biotechnology (CEMB), MHESI; Chulalongkorn University's Research Unit for Natural Product Biotechnology (Grant No. GRU6101133003-1) and by The National Research Council of Thailand (Grant No. N72B650340)”

“This work was supported financially by the PERDO’s Center of Excellence on Medical Biotechnology (CEMB). W.D. would like to thank Chulalongkorn University for providing some basic research fund for the Research Unit (Grant No. GRU 6101133003-1) and The National Research Council of Thailand (NRCT) (Grant No. N72B650340). S.N. would like to thank Chulalongkorn University for providing the King Bhumibol Adulyadej’s 72nd Birthday Anniversary Scholarship (72nd King’s Birthday Scholarship). WP would like to thank The Thailand Research Fund (TRF) for her Ph.D. Scholarship through RRi Program.”

“PERDO's Center of Excellence on Medical Biotechnology (CEMB), MHESI; Chulalongkorn University's Research Unit for Natural Product Biotechnology (Grant No. GRU6101133003-1) and by The National Research Council of Thailand (Grant No. N72B650340)”

7. PLOS ONE now requires that authors provide the original uncropped and unadjusted images underlying all blot or gel results reported in a submission’s figures or Supporting Information files. This policy and the journal’s other requirements for blot/gel reporting and figure preparation are described in detail at https://journals.plos.org/plosone/s/figures#loc-blot-and-gel-reporting-requirements and https://journals.plos.org/plosone/s/figures#loc-preparing-figures-from-image-files. When you submit your revised manuscript, please ensure that your figures adhere fully to these guidelines and provide the original underlying images for all blot or gel data reported in your submission. See the following link for instructions on providing the original image data: https://journals.plos.org/plosone/s/figures#loc-original-images-for-blots-and-gels.

Additional Editor Comments:

Dear Authors

I have a minor comment as it is interesting in the field.

It can be further considered when the minor brevision is made.

Reviewers' comments:

Reviewer's Responses to Questions

**Comments to the Author**

1. Is the manuscript technically sound, and do the data support the conclusions?

Reviewer #1: Partly

2. Has the statistical analysis been performed appropriately and rigorously? 

Reviewer #1: N/A

3. Have the authors made all data underlying the findings in their manuscript fully available?

Reviewer #1: Yes

4. Is the manuscript presented in an intelligible fashion and written in standard English?

Reviewer #1: No

5. Review Comments to the Author

Reviewer #1: This study is meaningful as a basic study for the development of a hair growth promoting mechanism and androgenic alopecia treatment of Avicennia marina extract and its active ingredient (avicequinone C).

We recommended that the following corrections are necessary.

1. There are no results regarding the cytotoxicity of AM and AC. It is necessary to write a separate Fig. In addition, it is necessary to description in detail why the experiment was performed only on low concentrations of the sample with AM (0-40 µg/mL) and AC (0-40 µM), and why the time was measured as 48 hours.

2. When measuring the protein expression of hair growth factor, the AM treatment concentration was effective at 10 µg/mL. When translocation from the cytoplasm to the nucleus was confirmed, the AM treatment concentration was increased by 2 times (20 µg/mL). What is reason?

3. In the cell cycle experiment, are there any results for each AM and AC treatment time period and concentration? What is the rationale for setting AM (10 µg/mL) and AC (10 µM) concentrations? And, what is the rationale for setting 48 hours?

4. Line 438: This may reflect the synergistic effects of the constituents present in the AM extract. → It is necessary to supplement the consideration of other active ingredients present in Avicennia marina.

5. Line 240: 366 font size

Line 248: Unit correction required (mM → µM)

Line 478: T?? DHT??

Line 528: in vitro, in vivo in italics

Line 638: 5 font size

Figure 3 (a) 3 Lane, unit correction required mg/mL → µg/mL

Figures 1, 5-8 Numbers and symbols overall poor readability. Needs correction

6. PLOS authors have the option to publish the peer review history of their article (what does this mean?). If published, this will include your full peer review and any attached files.

Reviewer #1: No

---

## [Author Response · Author response to Decision Letter 0]

10 Mar 2023

Response to Reviewers’ Comments

Manuscript: [PONE-D-22-22814] 

Title: Mechanistic synergy of hair growth promotion by the Avicennia marina extract and its active constituent (avicequinone C) in dermal papilla cells isolated from androgenic alopecia patients

Reviewers’ Comments

1. There are no results regarding the cytotoxicity of AM and AC. It is necessary to write a separate Fig. In addition, it is necessary to description in detail why the experiment was performed only on low concentrations of the sample with AM (0-40 µg/mL) and AC (0-40 µM), and why the time was measured as 48 hours.

Response: We have included the results of cytotoxicity of AM and AC as Fig. 2 in the main manuscript with describing the detail of the results and adding figure legend. For the question on the measuring time of 48 hours, this was obtained from the time-course study of 5α-R enzyme activity assay (0, 6, 12, 24, 36 and 48 h) using the primary DPCs (S6 Fig). The results showed that the activity of 5α-R enzyme was significantly increased during. Therefore, the cytotoxicity of AM and AC treatment was reasonably detected at 48h for cell viability. 

Revision:

Line 403-416: “In order to observe the effects of AM and AC on the cell cycle of the isolated DPCs, the primary cells were first determined for the cytotoxicity of both the extract and active compound by using MTT assay for 48 hr. This incubation time was optimized based on the time-course study of 5α-R enzyme activity assay (0, 6, 12, 24, 36 and 48 h) using the primary DPCs (S6 Fig). The results showed that the treatment of AM at the concentration range of 2.5-20 µg/mL and AC at 2.5-20 µM did not cause significant cytotoxicity on DPCs, compared with the non-treated cells (Fig 2). At the maximum non-toxic concentrations of 20 µg/mL AM and 20 µM AC, the cell viability was observed to be in the range of 80-85% which is lower than the acceptable 90% cell viability. Therefore, we selected the concentrations of 10 µg/mL AM and 10 µM AC for cell cycle analysis and further experiments.” 

Line 428-431: “Fig 2. Cytotoxicity assay of AM and AC on DPCs. DPCs were treated with various concentrations of (a) AM (0-40 µg/mL) and (b) AC (0-40 µM) for 48 h. Cell viability was evaluated by PrestoBlue® assay. The data is presented as mean + SD (n=3). * p < 0.05 versus non-treated cells.”

2. When measuring the protein expression of hair growth factor, the AM treatment concentration was effective at 10 µg/mL. When translocation from the cytoplasm to the nucleus was confirmed, the AM treatment concentration was increased by 2 times (20 µg/mL). What is the reason?

Response and Revision: It was a typo error. We are sorry for the mistake. In fact, the protein expression analysis of hair growth factors and translocation were all treated with AM at 10 µg/mL. We have corrected the concentration from 20 µg/mL to 10 µg/mL in the Line 324 and 513.

3. In the cell cycle experiment, are there any results for each AM and AC treatment time period and concentration? What is the rationale for setting AM (10 µg/mL) and AC (10 µM) concentrations? And, what is the rationale for setting 48 hours?

Response: For the cell cycle experiment, the concentration and time of AM and AC treatment were decided to set at 10 µg/mL and AC 10 µM which were based on the cytotoxicity result in Fig2. AM (10 µg/mL) and AC (10 µM) showed acceptable cell viability more than 90%. For the question on the measuring time of 48 hours, this was obtained from the time-course study of 5α-R enzyme activity assay (0, 6, 12, 24, 36 and 48 h) using the primary DPCs (S6 Fig). 

Line 736-737: S6 Fig. A TLC plate shows the optimization of 5α-R activity in various time points. DP primary cells were treated with substrate testosterone to detect the optimum time of dihydrotestosterone formation at 0-48h (lane 1-6). DP primary cells without testosterone (lane 7). Testosterone (lane8) and dihydrotestosterone (lane9).

Revision: Line 403-416: “In order to observe the effects of AM and AC on the cell cycle of the isolated DPCs, the primary cells were first determined for the cytotoxicity of both the extract and active compound by using MTT assay for 48 hr. This incubation time was optimized based on the time-course study of 5α-R enzyme activity assay (0, 6, 12, 24, 36 and 48 h) using the primary DPCs (S6 Fig). The results showed that the treatment of AM at the concentration range of 2.5-20 µg/mL and AC at 2.5-20 µM did not cause significant cytotoxicity on DPCs, compared with the non-treated cells (Fig 2). At the maximum non-toxic concentrations of 20 µg/mL AM and 20 µM AC, the cell viability was observed to be in the range of 80-85% which is lower than the acceptable 90% cell viability. Therefore, we selected the concentrations of 10 µg/mL AM and 10 µM AC for cell cycle analysis and further experiments.” 

4. Line 438: This may reflect the synergistic effects of the constituents present in the AM extract. → It is necessary to supplement the consideration of other active ingredients present in Avicennia marina.

Response This sentence intended to explain the observed results obtained from the AM extract. However, we agree that an additional sentence should be added to make it clearer.

Revision: Line 453-456: “This may reflect the synergistic effects of the constituents present in the AM extract [40]. To prove this, however, it is necessary to find other active constituents in AM, followed by observing the 5α-R inhibitory activity of their combinations with AC.” 

5. Line 240: 366 font size

Revision: We have revised 366 font size.

Line 248: Unit correction required (mM → µM)

Revision: We have corrected the unit from mM to µM 

Line 478: T?? DHT?? 

Revision: We have revised line 496-497 “These results suggest that the AM extract and the AC were able to inhibit both DHT-AR and T-AR complex formation.

Line 528: in vitro, in vivo in italics

Revision: We have revised italics font in line 545

Line 638: 5 font size

Revision: We have revised in line 655.

Figure 3 (a) 3 Lane, unit correction required mg/mL → µg/mL

Revision: The unit of mg/mL has been changed to µg/mL in Fig. 4

Figures 1, 5-8 Numbers and symbols overall poor readability. Needs correction.

Revision: The numbers and symbols were all corrected.

---

## [Editor Report · Decision Letter 1]

11 Apr 2023

Mechanistic synergy of hair growth promotion by the Avicennia marina extract and its active constituent (avicequinone C) in dermal papilla cells isolated from androgenic alopecia patients

PONE-D-22-22814R1

Dear Dr. De-Eknamkul,

We’re pleased to inform you that your manuscript has been judged scientifically suitable for publication and will be formally accepted for publication once it meets all outstanding technical requirements.

Kind regards,

Cheorl-Ho Kim, Ph.D.

Academic Editor

PLOS ONE

Additional Editor Comments (optional):

Dear Dr Wanchai

I have carefully read your revision and found it appropriately responded and revised, as you did.

I think that your study is interesting in the acting compound, avicequinone C, which was isolated from the Avicennia marina.

As the dermal papilla cells isolated from androgenic alopecia patients has been examined for hair growth promotion, the study will enahance our journal Plos One's impact.

I am very happy to decide editorial ACCEPT for publication in Plos One.

Thank you for your chosing the Plos One, once again,

Sincerely

Cheorl-Ho Kim PhD,

Editor, Plos One

Professor

SKKU, Biological Science

Professor, SHAIST.
---

## [Editor Report · Acceptance letter]

14 Apr 2023

PONE-D-22-22814R1 

Mechanistic synergy of hair growth promotion by the Avicennia marina extract and its active constituent (avicequinone C) in dermal papilla cells isolated from androgenic alopecia patients 

Dear Dr. De-Eknamkul:

I'm pleased to inform you that your manuscript has been deemed suitable for publication in PLOS ONE. Congratulations! Your manuscript is now with our production department. 

Kind regards, 

on behalf of

Professor Cheorl-Ho Kim 

Academic Editor

PLOS ONE